# Transition Metal Complexes of Thiosemicarbazides, Thiocarbohydrazides, and Their Corresponding Carbazones with Cu(I), Cu(II), Co(II), Ni(II), Pd(II), and Ag(I)—A Review

**DOI:** 10.3390/molecules28041808

**Published:** 2023-02-14

**Authors:** Ashraf A. Aly, Elham M. Abdallah, Salwa A. Ahmed, Mai M. Rabee, Stefan Bräse

**Affiliations:** 1Chemistry Department, Faculty of Science, El-Minia University, Minia 61519, Egypt; 2Institute of Organic Chemistry, Karlsruher Institut fur Technologie, 76131 Karlsruhe, Germany; 3Institute of Biological and Chemical Systems (IBCS-FMS), Karlsruhe Institute of Technology, Eggenstein Leopoldshafen, 76344 Karlsruhe, Germany

**Keywords:** thiosemicarbazides, thiosemicarbazones, synthesis, Cu(I), Cu(II), Co(II), Ni(II), Pd(II) and Ag(I) complexes, biological activity, catalysis

## Abstract

This review focuses on some interesting and recent applications of transition metals towards the complexation of thiosemicarbazides, thiocarbohydrazides, and their corresponding carbazones. We started the review with a description of the chosen five metals, including Cu[Cu(I), Cu(II], Co(II), Ni(II), Pd(II), and Ag(I) and their electronic configurations. The stability of the assigned complexes was also discussed. We shed light on different routes describing the synthesis of these ligands. We also reported on different examples of the synthesis of Cu(I), Cu(II), Co(II), Ni(II), Ag(I), and Pd(II) of thiosemicarbazide and thiocarbohydrazide complexes (until 2022). This review also deals with a summary of the fruitful use of metal complexes of thiosemicarbazones and thiocarbazones ligands in the field of catalysis. Finally, this recent review focuses on the applications of these complexes related to their biological importance.

## 1. Introduction

Metals complexes have been proven beneficial in drug development and medicinal chemistry [1,2]. Transition metals have partially filled d-orbitals and show variable oxidation states. Elements such as Cu(I), Cu(II], Co(II), Ni(II), Pd(II), and Ag(I) constitute transition metals or d-block elements. Their comparative stability in different oxidation states renders the metals a significant role in biological redox reactions [3]. Moreover, transition metal catalysts have become widely adopted as useful tools in modern synthetic organic chemistry because of their diverse reactivity in enabling various molecular transformations [4]. The chemistry has grown with the development of supporting ligands, which significantly affect the reactivity and stability of the metal complexes in the primary coordination sphere [4]. On the other side, thiosemicarbazones and thiocarbazones act as active ligands because of the following:i.They have better coordination tendencies.ii.They form more stable complexes.iii.They have better selectivity.iv.They may form macrocyclic ligands.

The Schiff bases of thiosemicarbazone and thiocarbazones display a wide range of coordination modes with transition metals; the number and the type of substituents affect the coordination mode [5,6].

### 1.1. Electronic Configuration of the Assigned Metals

The electronic configuration of transition metals shows them in many oxidation states. Although the elements can display many different oxidation states, they usually exhibit a common oxidation state depending on their most stable forms. In this study, we choose some selective metal ions named Cu(I), Cu(II), Co(II), Ni(II), Pd(II), and Ag(I) (Figure 1) that work only with the first row of transition metals.

Transition metal ions usually increase the biological activity of many ligands, and in certain cases, the activity has been attributed entirely to the corresponding metal ions [7]. Studies have proven that metal complexes exhibit a wide range of biological and chemical characteristics when bound with organic moieties [8,9]. Due to the change in the geometric properties of the organic molecule–metal binding compared with organic moiety itself, the structural and electronic properties of transition metals would consequently be changed [10,11]. Transition metals react with various negatively charged molecules due to their various oxidation states [12]. The combination between these transition metals and organic moieties can be directed towards a certain geometry for interactions with biological targets [10]. That combination can also occur between metals and organic compounds containing nitrogen heteroatoms, which are electronically shown in Figure 2.

### 1.2. Stability of Transition Metal Complexes

The stability of the transition metal complexes depends on various factors, such as the crystal field stabilization energy (CFSE), the attainment of effective atomic number, and the chelation/denticity of the ligands. The CFSE is the difference in energy gap between the eg (the upper part with higher energy) and the t2g (the lower part with lower energy) orbitals. The octahedral complexes are more stable than tetrahedral complexes because they are surrounded by six ligands causing more electronic repulsion and hence high splitting energy. As the CFSE increases, the thermodynamic stability, especially of the octahedral complexes, increases [13].

### 1.3. Metal Complexes of Copper (Cu)

Copper is an important micronutrient for all species that live in oxygen-rich environments. It is a redox-active metal that rapidly transitions from the reduced Cu(I) oxidation state to the oxidized Cu(II) oxidation state or vice versa in both traditional bench chemical processes and physiological settings [14]. The Cu(I) ions have a d^10^ configuration in compounds, and therefore, they become diamagnetic [15]. The oxidation state of the d elements has been linked to a considerable increase or decrease in the stability of their corresponding complexes [14]. All copper (II) complexes are paramagnetic by nature due to unpaired electrons [16,17].

### 1.4. Metal Complexes of Cobalt (II)

There are 27 essential elements for maintaining and growing the human body, out of which cobalt(II) is one of the most important and essential elements [18]. Co(II) complexes have been studied and synthesized widely because of their various properties [19]. Cobalt is an essential trace element found in all animals and is employed as a cofactor of vitamin B12; consequently, it can regulate the synthesis of DNA and maintain the normal functioning of the nervous system and brain [20]. The synthesis and reactivity of cobalt complexes of Schiff base ligands have always attracted the attention of inorganic chemists [21,22]. For instance, the cobalt complexes of tetradentate Schiff base ligands have been extensively used to mimic cobalamin (B12) coenzymes [23,24], dioxygen carriers, and oxygen activators [25,26,27]. They are also used for enantioselective reduction [28] and as antimicrobial agents [29].

### 1.5. Metal Complexes of Nickel(II)

Nickel plays an important role in the biology of microorganisms and plants [30]. One of the carbon monoxide dehydrogenase enzymes consists of the Fe-Ni-S cluster [31]. Another nickel-containing enzyme found in the rare bacterial class is superoxide dismutase [32]. Nickel(II) Schiff base complexes containing sulphur donors have received attention due to identifying a sulphur-rich coordination environment in biological nickel centres, such as the active sites of certain ureases, methyl-*S*-coenzyme-*M*-methyl reductase, and hydrogenases [33,34].

### 1.6. Metal Complexes of Palladium(II)

Palladium is a d-block element having atomic number 46. The group number and period numbers of palladium are 10 and 5, respectively. Therefore, the electronic configuration is [Kr] 4d^10^. Palladium is also an alternative that has shown considerable promise in developing metal-based anti-cancer drugs [35,36,37]. Several similar results have been observed between platinum (II) and palladium (II) complexes, which were thought to be anti-cancer agent candidates [38,39]. Therefore, researchers have focused on synthesizing new palladium complexes to treat several cancer types [40]. The thiosemicarbazone and thiocarbazone derivatives of Pd(II) have proven to be more effective as anti-cancer or antimicrobial agents than the ligand, probably due to the increased lipophilicity of the complexes compared to the free ligands alone [41].

### 1.7. Metal Complexes of Silver (I)

Silver-based compounds have been explored as drug candidates for cancer chemotherapy [42]. Structural membrane alteration, enzyme inactivation by interaction with nucleic acids, and induction of oxidative stress have been frequently associated with their toxic effects. The cytotoxicity of silver complexes to cancer cells strongly depends on lipophilicity, solubility, and stability in aqueous media [42,43]. Chelation is a well-known strategy to modulate the physicochemical properties of silver compounds and to incorporate ligands that are biologically active by themselves [40,42]. Metal complexes of silver (Ag) have good antibacterial activity, and silver-based antimicrobials are attractive in terms of their effectiveness and non-toxic behaviour to human cells [3].

Silver is one of the biological metals found in our bodies in trace amounts [44]. Notably, the binuclear silver(I) complexes exhibited a more significant antiproliferation activity in cancer cells than the normal cells and exhibited low toxicity [44]. The closed d^10^ valence shell of silver(I) tolerates various coordination geometries [45]. Silver(I) complexes can be either two-coordinate and approximately linear or three- or four-coordinate [46], but the use of chelating ligands often leads to polymeric species or multinuclear clusters [47]. The coordination chemistry of transition Ag with thiosemicarbazones is broadly investigated [48], but within this realm, only a few compounds of silver(I) thiosemicarbazone complexes have been reported [49].

Based upon those above, this review showed some representative examples of recent synthetic tools of thiosemicarbazones. We also illustrated some spots on the ligation of the assigned metals towards thiosemicarbazones and thiocarbazones. Moreover, we gave brief notes about the utility of the assigned metal-thiosemicarbazones in the field of catalysis. Then, we dealt with the biological activities of the formed metal complexes.

## 2. Synthesis of Thiosemicarbazide Derivatives

The synthesis of thiosemicarbazides and/or thiocarbazones may be carried out in several ways.

### 2.1. Using Carbon Disulphide

The reaction of arylamines with carbon disulphide (CS_2_) in the presence of KOH gave potassium arylcarbamodithioates **1**, which when treated with methyl iodide, the reaction produced *N*-arylmethyl-carbamodithioates **2**. The target 4-arylthiosemicarbazides **3a**–**e** were obtained during the reaction of **2** with hydrazine, as shown in Figure 1 [50].

### 2.2. Using Ammonium Thiocyanate

1-Aroylthiosemicarbazides **5a**–**f** were obtained via the reaction of aroyl hydrazines **4a**–**f** with ammonium thiocyanate in acetone (Figure 2) [51].

### 2.3. Reactions of Hydrazines with Substituted Isothiocyanate

Recently, Aly et al. reported that 2-(4′-[2.2]paracyclophanyl-4*H*-*N*-substituted-hydrazine-carbothioamides **7a**–**f** were prepared in 80–88% yields by refluxing compound **6** with the isothiocyanates in EtOH for 6–8 h (Figure 3) [52].

Previously, it was reported that treatment of ethyl benzoate (**8**) with hydrazine, the corresponding benzoyl hydrazine (**9**), obtained an 85% yield (Figure 4). Subsequently, on subjecting **9** with ammonium thiocyanate in an acidic medium, *N*-benzoylthiosemicarbazide (**10**) was formed in 75% yield (Figure 4) [53].

### 2.4. Representative Examples of Synthesized Thiosemicarbazones

The reaction of 2-hydroxy benzaldehyde with thiosemicarbazide (**11a**) in refluxing EtOH and in the presence of Na_2_CO_3_ afforded 2-hydroxybenzaldehyde thiosemicarbazone (**12**) in 40% yield (Figure 5) [54], whereas when refluxing 5-hydroxy-2-nitrobenzaldehyde (**13b**) with thiosemicarbazide (**11a**) in ethanol, the reaction afforded (*E*)-2-(5-hydroxy-2-nitrobenzylidene)-hydrazine-1-carbothioamide (**14**) in 84% yield (Figure 5) [55].

Tada et al. reported that thiosemicarbazone **15** was obtained, in 80% yield, during the reaction of 3-bromo-4-hydroxy-5-methoxybenzaldehyde with 4-(4-bromophenyl)-thiosemicarbazide (**11b**) in MeOH and catalysed with few drops of glacial acetic acid (Figure 6) [56].

An interesting approach reported that (*E*)-2-((4-chlorophenyl)(2-hydroxy-phenyl)methylene)-hydrazine-carbothioamide (**16**) was obtained, in 70% yield, via a three-component one-pot reaction between phenol, 4-chlorobenzaldehyde, and **11a** (Figure 7). The reaction was performed in refluxing EtOH for 6 h (Figure 7) [57]. All the test compounds were screened for in vitro antibacterial activity against *B. subtilis*, *S. aureus*, and *S. Typhi* and showed significant antibacterial activity against the bacterial strains used. The values compare well with the potency of Ampicillin in the respective assay [57].

When 1-(4-hydroxy-6-methyl-4*H*-pyran-2-yl)ethan-1-one (**17**) was led to react with phenyl-thiosemicarbazide (**11c**) in refluxing methanol for 3 h, the reaction yielded the corresponding thiosemicarbazone having pyrano group **18** in 78% yield (Figure 8) [58].

In the same manner, the reaction of 4-(3′,4′-benzaldehyldene)-2,3-dimethyl-1-phenyl-3-pyrazolin-5-ones **19a,b** with **11a** in refluxed EtOH for 4–5 h afforded the corresponding Schiff bases **20a,b** in 65–80% yield (Figure 9) [59].

Bis-thiosemicarbazone derived by benzil **22** was prepared, in 65% yield, during the reaction of two equivalents of thiosemicarbazide **11a** with one equivalent of benzil (**21**) in EtOH and an acidic medium (Figure 10) [60]. Compound **22** was tested in vitro against several microorganisms to assess their antimicrobial properties and showed moderate activity [60].

The isatin-based thiosemicarbazones **24a**–**c** were synthesized in 85–90% yields by the condensation between substituted isatin derivatives **23a**–**c** and 4-phenyl thiosemicarbazide (**11c**) in the presence of acetic acid under heating at 70 °C (Figure 11) [61].

The synthetic approach of thiosemicarbazones containing [2.2] paracyclophane moiety was described during the condensation reactions between **25a**–**c** and thiosemicarbazides **11a**,**c,d** to yield **26a**–**e** (Figure 12). The reaction proceeded in refluxing EtOH and in the presence of a few drops of conc HCl. Treatment of two equivalents of 4-formyl-[2.2]paracyclophane (**25a**) with one equivalent of thiocarbohydrazide (**11d**) in ethanol and catalysed with Et_3_N, gave bis-thiocarbohydrazone **27** in 90% yield (Figure 12) [62].

Thiosemicarbazones derived by 1,4-naphthoquinone **29a**–**f** were established upon mixing equimolar amounts of thiosemicarbazides **11e**–**j** with 2,3-dichloro-1,4-naphthoquinone (**28**), Ph_3_P and in the presence of Et_3_N as a catalyst and acetonitrile as a solvent. Triphenylphosphine-ylidene)-3,4-dihydronaphthalen-2(1*H*)-ylidene)-*N*-substituted-hydrazine-1-carbothioamides **29a**–**f** were obtained in 72–82% yield (Figure 13) [63].

## 3. Metal Complexes of Thiosemicarbazones and Thiocarbazones

Thiosemicarbazones and thiocarbazones usually bind to a metal ion as bidentate N,S-donor ligands, forming five-membered chelate rings [64]. When a third donor site (D) is incorporated into the ligands, normally D,N,S–tri-coordination occurs [65]. They are used as chelating ligands for forming metal complexes because of the various flexible donor sets of sulphur and nitrogen [66,67]. Coordination chemistry of Schiff base ligands is a promising research area in modern chemistry, as metal complexes play important roles in biological systems. The chemical and biological importance of metal complexes containing a Schiff base depends on the presence of an azomethine (C = N) linkage [68].

Further, the sulphur-containing ligands are known for their biocidal activities, owing to their ability to chelate with the soft acidic metal ions at the tracer level. Among organic reagents containing S and N as donor atoms, thiosemicarbazones and their aromatic derivatives occupy a unique place [69]. Thiosemicarbazones and thiocarbazones can coordinate as a neutral, bidentate NS chelate or, more commonly, as an anionic bidentate NS chelate upon deprotonation of the azomethine nitrogen [70].

### 3.1. Cu Complexes of Thiosemicarbazides and Thiosemicarbazones

Tetradentate Cu complexes **31a**–**f** were obtained in good yields by adding *N*-substituted thiosemicarbazones **30** with copper(I)salts in acetonitrile (CH_3_CN). When the reaction was continued with stirring for 3 h, a yellow precipitate was formed. The subsequent addition of triphenylphosphine (Ph_3_P) yielded the corresponding **31a**–**f** (Figure 14) [71].

In the same manner, bidentate copper(I)-9-anthraldehyde thiosemicarbazone complexes **33a,b** were obtained by refluxing of 9-anthraldehyde thiosemicarbazone (**32**) with Ph_3_P and CuX in a (1:1:1) molar ratio using acetonitrile as a solvent under reflux for 24 h. The reaction gave metal complexes good yields (Figure 15) [72]. The interactions of the ligands and their Cu(I) complexes with calf thymus DNA (ct-DNA) and human serum albumin (HSA) were examined through UV–visible and fluorescence spectroscopy. Results showed that one copper(I) complex displayed strong interactions with ct-DNA and HSA, having binding constant values of 6.66 × 10^4^ and 3.28 × 10^4^ M^−1^, respectively. The complexes also showed good interactions with DNA in docking studies [72].

The Cu–S bond distance of 2.3942 Å in 33a is close to those of other monomeric complexes of copper(i) with thiosemicarbazones. The Cu–Cl bond distance of 2.3754(6) Å is smaller than the sum of ionic radii of Cu^+^ and Cl^−^ (2.58 Å). The other bond distances are Cu–S (2.3942(5) Å) and Cu–P (2.2732(4), 2.2839(4) Å). The C−S distance of 1.694(2) Å is longer than C = S (1.62 Å) and smaller than a C−S single bond (1.81 Å), indicating its partial double bond character in the metal complex [72]. The bond angles around the copper atom in the range of ca. 102.06(2)°–109.65(2)° reveal a distorted tetrahedral geometry with maximum distortion in the P–Cu–P bond angle (124.99(2)°) [72].

Similarly, the reaction of *N*-substituted 2-tosyl-diazane-1-carbothioamides **34a,b** with the [CuCl(PPh_3_)_3_] in a 1:1 ratio in ethanol under reflux for 2 h afforded 74–87% of *N*-substituted-2-tosyl-diazane-1-carbothioamide copper complexes (cyclohexyl and phenyl) **35a,b** (Figure 16) [73]. The crystal structures confirmed the tetrahedral geometry for the Cu(I) metal centre, coordinated by one chlorine atom, two triphenylphosphane, and by the thiosemicarbazide as a neutral *S*-donor monodentate ligand [73].

Photophysical studies revealed that the complexes exhibit emission at room temperature with maxima around 480 nm. At 77 K, the emission is shifted to higher energy, a characteristic behaviour of MLCT (metal-to-ligand charge transfer) emitters. The experimental and theoretical analyses suggest ^3^MLCT (triplet state of metal-to-ligand charge transfer, d-π∗ character) radiative decay occurrence for the complexes. Furthermore, biological assays showed that the complexes are active on the intracellular amastigote form of *Trypanosoma cruzi* (Tulahuen Lac-Z strain) and present a significant cytotoxic effect against metastatic melanoma cells [73].

It was reported that (*E*)-2-(tert-butyl)-*N*,*N*-diethyldiazene-1-carbothioamide (**36**) formed with monovalent copper salt in THF under reflux, the tetradentate copper complexes **37a,b** in 55–62%, whereas with [(CH_3_CN)_4_Cu]BF_4_ salt, Cu(I)-complex **38** was obtained in 69% yield (Figure 17) [74].

Another bidentate metal complex structure of Cu(II) was found during the reaction of (*E*)-2-(3,4-disubstituted-benzylidene)-*N*-methylhydrazine-1-carbothioamides **39a,b** with copper(II) chloride in EtOH under reflux to give the bidentate copper complexes **40a,b** in good yields (Figure 18) [75]. The structure–activity relationship indicated that one complex exhibited a significant effect against *E. faecalis* and *A. baumanii*, but others acted only against *E. faecalis*. In addition, all the complexes showed significant antimicrobial activities against Gram-positive and Gram-negative bacteria and two human fungal pathogens [75]. The ligands **39a,b** and their complexes **40a,b** revealed greater and more specific activities against the studied bacterial and fungal strains. Therefore, it was indicated that the complexes of thiosemicarbazones and their derivatives are the most extensively considered compounds due to their potential therapeutic use as antifungal, antiviral, or antibacterial agents [75].

Moreover, when an ethanolic solution of CuCl_2_.2H_2_O was added to an ethanolic solution of (*E*)-2-(2,4-dihydroxybenzylidene)-*N*-methyl-*N*-phenylhydrazine-1-carbothioamide (**41**) and the reaction was heated under reflux for 4 h, the reaction proceeded to give tetradentate complexes **42a,b** in 61 and 63% yield, respectively (Figure 19) [76].

Electronic and vibrational absorption spectra of the nickel complex were found to be of square-planar geometry. The thermogravimetric analyses of these complexes confirmed the presence of water molecules in their structures, and thermal decomposition led to the formation of metal oxides as the latest residues. The Ni(II) complex’s voltammogram suggests a quasi-reversible redox system in DMSO solution [76].

The reaction of equivalent amounts of 2-(thiophen-2-ylmethylene)hydrazine-1-carbothioamide (**43**) with an ethanolic solution CuCl_2_.2H_2_O was stirred and heated under reflux for 3 h (Figure 20) and produced pale-yellow powder of metal tetradentate complexes **44** in nearly 60% yield [77]. The as-prepared complex was used as single-source precursors for the synthesis of oleylamine-capped (OLA@CuxSy), hexadecyl amine-capped (HDA@CuxSy), and dodecylamine-capped (DDA@CuxSy) copper sulphide nanoparticles (NPs). This study confirmed the potential of synthesized copper sulphide nano photo-catalysts in treating water contaminated with organic pollutants [77].

Copper (II) thiosemicarbazones **46** (known as 5-nitroisatin-4-(1-(2-pyridyl)piperazinyl)-3-thiosemicarbazones (Nitistpyrdlpz) and their Cu(II) complex) were prepared by mixing CuCl_2_.2H_2_O and thiosemicarbazones **45** in absolute ethanol at 80 °C for 6 h. The reaction afforded **46** good yields (Figure 21) [78]. The synthesized compounds were tested against breast cancer cell lines, MCF-7, MDA-MB-231, and epidermoid carcinoma; A431 showed that the complex reduced cell viability percentage towards all the tested cell lines but made a remarkable contribution towards the MDA-MB-231 cell line. The IC_50_ of the complex and free ligand was found in the range of IC_50_ 0.85–1.24 μM and IC_50_ 3.28–3.53 μM, respectively. Among those cell lines, the complex may be the better anti-cancer agent for MDA-MB-231 because of its action at a micromolar concentration (IC_50_ 0.85 μM) [78].

The tetradentate complexes of Cu-thiosemicarbazones were formed during the reaction of thiosemicarbazones **47a**–**k** with CuCl_2_·2H_2_O in refluxing EtOH. The reaction afforded the copper complexes **48a**–**k** in 70–83% yields (Figure 22) [79]. All copper complexes were characterized by similar TGA thermograms exhibiting thermal stability up to ca 170 °C and decomposition pattern in five steps. For example, the TGA curve of Cu-Ligand·H_2_O demonstrated the first step in the range of 180–195 °C (mass loss 3.3%), indicating water elimination. The second step in the range of 196–275 °C (12.1%) could be due to the breakdown of the methoxy moieties. In the third step, phenyl azo moiety was lost at 275–446 °C (19.5%). The fourth step, recorded at 450–650 °C (48.8%), was attributed to the complete decomposition of the remaining organic components. The final step at ca. 650 °C indicated the decomposition of CuO [79].

A similar observation was found in the case of the formation of tetradentate Cu metal complex during the reaction of 2,4-pentanedione bis-thiosemicarbazone **49** with Cu(II) acetate under aerobic conditions to yield the mononuclear Cu-complex **50** (Figure 23) [80]. The methodology was established via stirring of Cu(CH_3_COO)_2_·H_2_O to the title thiosemicarbazone in acetone for 15 min at room temperature. The reaction produced dark brown crystals of copper complex **50** in 65% yield (Figure 23) [80].

Similarly, nonsymmetric tetradentate copper bis-thiosemicarbazone complexes **52a,b** were obtained by refluxing equimolar amounts of ligands **51a,b** with copper acetate hydrate in refluxing ethanol (Figure 24) [81].

Copper(I) formed complexes formed the tetradentate **52** and hexadentate **53** structures with thiosemicarbazones derived by [2.2] paracyclophane **25a,b**, as shown in Figure 25 [62].

Recently Aly et al. [63] reported that the reaction of equal equivalents of compounds **29a**–**f** with Ph_3_P and CuCl_2_.2H_2_O in ethanol at rt and for 1–3 d produced the bidentate Cu-complexes **54a**–**f** in 80–93% yields as indicated in Figure 26.

Another recent approach has dealt with the complexation of thiosemicarbazones **55a**–**k** with Cu(I) (Figure 26). The mixture was stirred in DMF for 2d, and bidentate metal complexes **56a**–**k** were obtained in 64–80% yields (Figure 27) [82].

Differently, the tridentate metal complexes **58a**–**c** were formed by mixing equimolar amounts of **57a,b** with Cu(I) salts (Br and I) in CH_3_CN. As the mixture was stirred for 24 h, metal complexes **58a**–**c** were obtained in 45–60% yields (Figure 28) [83].

### 3.2. Co(II) Complexes of Thiosemicarbazides and Thiosemicarbazones

The reaction of cobalt (II) chlorides and bromides with 3-thiophene aldehyde thiosemicarbazone **43** leads to the formation of complex **59** in 44% yield (Figure 27). The structure of **59** was elucidated as bis-(bi-dentate) of Co(II) with two molecules of the thiosemicarbazone **43** (Figure 29) [84].

The tetradentate Co-complex **61** was obtained by mixing an equivalent amount of (*Z*)-*N*-(4-methoxyphenyl)-*N*’-((*Z*)-propylidene)carbamoydrazonothioic acid (**59**) with Co(NO_3_)_2_.6H_2_O in ethanol under reflux for 3 h. After adding 30% of NH3 and stirring the mixture for 16 h, brown crystals of cobalt complex **61** were obtained in 53% yield (Figure 30) [85]. Compound **60** was suggested as the intermediate step before adding NH_3_ (Figure 30). Compound **61** was utilized as an electrocatalyst for the oxidation of hydrazine using a modified electrode approach [85].

Electrochemical studies such as cyclic voltammetry, linear sweep voltammetry, and chronoamperometry demonstrate superior electrocatalytic behaviour of the prepared complex compared to bare electrodes. Based on Tafel plot analysis, for the electrooxidation of hydrazine, the initial one electron transfer is found to be the rate-limiting step followed by fast three-electron transfers for complete oxidation to nitrogen. The chronoamperometry technique shows a selective response towards hydrazine electrooxidation in the presence of interfering agents and is sensitive with a detection limit of 1.7 μM. Accordingly, a cobalt complex-modified electrode could be used as an alternative electrocatalyst compared to a precious-metal-based electrocatalyst for hydrazine oxidation [85].

Surprisingly, adding an ethanolic solution of sodium acetate to an ethanolic solution of CoCl_2_.6H_2_O and thiosemicarbazone **62,** under refluxing conditions, gave hexadentate metal complexes **63** (Figure 31) [86].

### 3.3. Ni (II) Complexes of Thiosemicarbazides and Thiosemicarbazones

Ni(II) salts have shown that complexation of fluorene-2-carboxaldehyde thiosemicarbazides **64a**–**c** with Ni(OCOCH_3_)_2_ in refluxing ethanol for 4 h produced yellow crystals of tetradentate (bis-bidentate) nickel complexes **65a**–**c** in 40%, 49% and 50% yields (Figure 32) [87]. The results suggested that the thiosemicarbazone ligands behaved as bidentate ligands coordinated to the Ni(II) ion via their N and S atoms. Among the six tested compounds, two nickel complexes, 62b and 62c, exhibited moderate in vitro antimalarial activity with IC_50_ of 23.79 and 2.29 μM, respectively. As the size of the substituent group increases, the antimalarial activity of the compound increases. Complex 62c exhibited the highest antimalarial activity. In addition, ligands 61c and complex 62a showed higher cytotoxic activity against HCT 116 human colorectal carcinoma cell line than cisplatin with IC_50_ of 0.69 and 3.36 μM, respectively [87].

Mixed complexation was established when an ethanolic solution of thiosemicarbazone ligand **66a**–**d** was added to nickel(II)chloride hexahydrate and PPh_3_ and then the mixture was refluxed for 6 h to produce tetradentate Ni(II) thiosemicarbazone complexes **67a**–**d** in 63–71% as illustrated in Figure 33 [88]. The complexes were further characterized with single crystal X-ray diffraction. The complexes are four-coordinated and adopt a square planar geometry, in which the Schiff base ligands bind to the metal centre via their tridentate O, N, and S atoms. Complexes **67b** and **67c** showed moderate in vitro antimalarial activity with IC_50_ 9.88 ± 0.23 and 1.06 ± 0.01 μM, respectively. Remarkably, the antimalarial activity increases as the hydrophobicity of the substituent group attached at the N(3) position increases [88].

An interesting approach described the reaction of pyridine-2-carbaldehyde thiosemicarbazone (**68**) with Ni(NO_3_)_2_⋅6(H_2_O). Two isolated structures of hexadentate Ni(II) complexes **69a,b** were formed. The procedure depending upon mixing an equal equivalent of **68** and Ni(II) salt in EtOH at 50 °C for 1 h led to the formation of ethanol soluble red complex **69a** and water-soluble brownish complex **69b** (Figure 34) [89]. X-ray structure analysis showed that the packing structure of all Ni(II) cationic complexes, nitrates anions, and water molecules are linked to each other via N–H⋯O and O–H⋯O hydrogen bonds into a three-dimensional supramolecular network [89].

Hosseini-Yazdi et al. [90] prepared Ni(II), Cu(II) complexes **71a,b** by mixing equimolar amounts of Ni(COOCH3)2 with thiosemicarbazone **70a,b** under methanolic reflux for 4 h (Figure 35). The reaction gave complex **71a,b** in 72–79% yields, as shown in .

Ni-thiosemicarbazone complex **73** was obtained during heating of (*E*)-2-(1-(4-hydroxyphenyl)ethylidene)-*N*-(pyridin-2-yl)hydrazinecarbothioamide (**72**) with Ni(II) chloride under refluxing ethanol for 2 h. The naming complex was identified as (*E*)-2-(1-(4-hydroxyphenyl)ethylidene)-*N*-(pyridin-2-yl)hydrazinecarbothioamide Ni-complex **73** and was obtained in 85% yield as shown in Figure 36 [91]. The ligand behaves as NS-monodentate proposed square planner geometry with the most fitted one for the metal complex motif [91].

The most indicative appeared in the syntheses and characterization of some mixed-ligand nickel(II) complexes **75a**–**c** and **76a**–**c** of three selected thiosemicarbazones **74a**–**c** (Figure 37), which was previously reported [92]. The complexes have been screened for their antibacterial activity against *Escherichia coli* and *Bacillus* [92]. The results indicated that the corresponding nickel(II) complexes showed much better antibacterial activity concerning the individual ligands against the same microorganism under identical experimental conditions.

Selvamurugan et al. [93] synthesized three Ni(II) complexes of 4-chromone-4-substituted-thiosemicarbazone **78a**–**c** by refluxing ethanolic solution of 4-chromone-*N*(4)-substituted thiosemicarbazone **77a**–**c** with [Ni(NO_3_)_2_·6H_2_O] for 6h. A dark-green-coloured crystalline powder was obtained on slow evaporation (Figure 38). The complex’s single crystal X-ray diffraction study revealed a distorted octahedral geometry around the metal centre. The assigned complexes were subjected to biological investigations such as DNA/protein interaction studies and in vitro cytotoxic studies against human breast cancer cell lines (MCF-7). The DNA binding by fluorescence spectral study showed that the complexes bind to DNA via intercalation binding mode. Protein binding studies using fluorescence spectroscopy showed that the new complexes could bind strongly with bovine serum albumin (BSA). Due to the terminal substituted thiosemicarbazones, the complexes have good anti-cancer activity against the MCF-7 cancer cell line [93].

Bis-bidentate of Ni(II) complexes **80a**–**c** were synthesized in 67–83% yields by refluxing a mixture of hot ethanolic solutions of NiCl_2_.2H_2_O and appropriate ligands **79a**–**f** for 2 h (Figure 39) [94].

The addition of *S*-propyl-*N*-amino-3,5-dibromobenzylidene thiosemicarbazone (**81**) to 2-amino-3,5-dibromo benzaldehyde with NiCl_2_·6H_2_O in EtOH was achieved (Figure 40), to give brown crystals of complex **82** in 38% yield [95].

During heating, a methanolic solution of nickel(II) acetate tetrahydrate salt with (*E*)-*N*-methyl-2-(quinolin-2-ylmethylene)hydrazine-1-carbothioamide (**83**) dissolved in hot acetonitrile under stirring for 24 h at room temperature led to the formation of compounds bis-tridentate (hexadentate) nickel complex **84** in 84% yield (Figure 41) [96].

A report above [82] described the complexation of Cu(II) complexes with compounds **55a**–**k**, Ni(II) salts reacted with **55a**–**k** in refluxing CH_3_OH to undergo similar coordination, and the corresponding bidentate metal complexes **85a**–**k** were formed (Figure 42) in good to excellent yields [82].

### 3.4. Cu(II), Co(II) and Ni (II) Complexes of Thiosemicarbazides and Thiosemicarbazones

2-butyl thioquinazoline-4-(3*H*)-thiosemicarbazone (**86**) was added to Co(II), Ni(II), and Cu(II) salts in refluxing ethanol reflux for 3–4 h, and the reaction afforded 2-butyl thioquinazoline-4-(3*H*)-thiosemicarbazone complexes **87a**–**h** in 60–65% yields [97] (Figure 43). These complexes were also subjected to study their antimicrobial screening against, Gram-positive bacteria *Candida albicans* and Gram-negative bacteria *Escherichia coli* by disc diffusion technique [97].

However, the hexadentate metal complexes **88a**–**h** were formed during the mixing of two equivalents of **57a**–**f** with one equivalent of Cu(I) and Ni salts in refluxing CH_3_OH (Figure 44) [83]. Docking results supported the tested complexes’ ability to potentially inhibit the RdRp of SARS-COV-2 from showing binding energy higher than their corresponding ligands. Additionally, ADMET prediction revealed that some compounds stratify to Lipinski’s rule, indicating good oral absorption, high bioavailability, permeability, and transport via biological membranes. Therefore, these metal-based complexes are suggested to be potentially good candidates as anti-COVID-19 agents [83].

3-Chlorovanillin thiosemicarbazone metal complexes **90a**–**c** were synthesized by stirring metal acetates (Cu(II), Co(II), and Ni(II)) with the ligand of 3-chlorovanilin thiosemicarbazone (**89**) in dioxane resulting in good yields (72–76%) as illustrated in Figure 45 [98].

It was reported that some Schiff base ligand 2,6-pyridinedicarboxaldehyde-thiosemicarbazone (**91**) metal-complexes of Cu(II) **92a**, Co(II) **92b,** and Ni(II) **92c** had been synthesized using conventional as well as microwave methods (Figure 46) [99]. Microwave-assisted synthesis proved more efficient in terms of reaction time and availability of the product with better yield than conventional synthesis. Moreover, all the complexes were found to be semiconductors as per electrical conductivity data. Schiff base ligands and their complexes were also evaluated for their antimicrobial activity on selected bacteria, *E. coli* and *S. aureus*, and two fungi, *Aspergillus niger* and *Candida albicans*. Metal complexes were found to be more potent than the parent ligand molecule due to chelation, which makes the ligand act as a more powerful and potent bactericidal agent [99].

Transition metal complexes of Cu(II), Co(II), and Ni(II) **94a**–**c** were prepared by refluxing an ethanolic solution of the thiosemicarbazone ligand **93** with 1,10-phenanthroline heterocyclic base with the metal salts (Figure 4). The reaction mixtures were refluxed on a water bath for 4 h and gave 85, 82, and 79% for Cu(II), Ni(II), and Co(II) complexes, respectively (Figure 47) [100]. The magnetic and spectral data indicate octahedral structures for all complexes. Moreover, the free ligand and its M(II)-chelates have been screened for antimicrobial activity. The antibacterial screening demonstrated that the Cu(II) complex has the maximum as well as broad activities among the investigated complexes [100]. The comparison of the antimicrobial activities of the compounds against the selected types of microorganisms indicates that Cu(II) > Ni(II) > Co(II) [100].

The reaction of (*Z*)-2-(pyrrolidin-2-ylidene)hydrazine-1-carbothioamide (**95**) with metal salts in refluxing ethanol for 4 h gave (*Z*)-2-(pyrrolidin-2-ylidene)hydrazine-1-carbothioamide complexes **96a**–**c** in 75%, 60% and 77% yields for Co(II), Ni(II) and Cu(II) complexes, respectively (Figure 48) [101]. All complexes were found to be superior antioxidants compared to ascorbic acid. In addition, in vitro antifungal effects of the investigated compounds were tested against two fungal species (*Aspergillus niger* and *Candida albicans*). The results showed that the ligand does not exhibit antifungal activity, but all metal–ligand complexes exhibit good activities [101].

When vitamin K3 thiosemicarbazone **97** was dissolved in aqueous EtOH, and the solution was slowly added with stirring into a freshly prepared solution of the metal chloride (Cu(II), Co(II), and Ni(II) in EtOH, the reaction proceeded to give thiosemicarbazone metal complexes **98a**–**c** as illustrated in Figure 49 [102]. All of the complexes possess strong inhibitory action against G(+) *Staphylococcus aureus*, G(−) *Hay bacillus*, and G(−) *Escherichia coli*. The antibacterial activities of the complexes are stronger than those of the thiosemicarbazone itself. The antibacterial effect of the nickel(II) complex was similar to that of penicillin against the two G(+) strains [102].

Additionally, cobalt and nickel nitrates formed with thiosemicarbazone of glyoxylic acid **99** metal complexes **100a,b** via stirring a mixture of an aqueous solution of thiosemicarbazone ligand with either NiCl_2_ or Co(NO_3_)_2_·4H_2_O for 5 min at room temperature, resulting in 82 and 86% yields for Ni(II) and Co(II) complexes, respectively (Figure 50) [103]. It was shown that the thiosemicarbazone of glyoxylic acid metal derivatives had effective inhibition against α-glycosidase, cytosolic carbonic anhydrase I and II isoenzymes, butyrylcholinesterase, and acetylcholinesterase. *K*_i_ values were 26.12–36.58 nM for hCA I, 20.73–40.78 nM for hCA II, 184.30–642.18 nM for AChE, 123.67–342.37 nM for BChE, and 14.66–45.62 nM for α-glycosidase [103].

In 2021, V. K. Revankar et al. [104] reported that 8-hydroxyquinoline derived *p*-halo *N*^4^-phenyl substituted thiosemicarbazones **101a**–**c** (i.e., prepared from a reaction of 2-formyl-8-hydroxyquinolines **102** with aryl thiosemicarbazides **11b,k**,**l**) formed various structures of metal complexes **103a**–**c** and **104a**–**c** with Cu(III), Ni(II) and Co(II), respectively (Figure 51). The various physicochemical investigations of the synthesized complexes reveal metal to ligand stoichiometry to be 1:2 in Co(III) complexes and 1:1 in Ni(II) and Cu(II) complexes. The ligands coordinate in a tridentate NNS fashion around Co(III) centres to form an octahedral geometry and square planar geometry around Ni(II) and Cu(II) metal centres. Co(II) oxidation to Co(III) was also observed during complexation. The synthesized compounds are subjected to in vitro cytotoxicity studies. Compared to bare ligands, the complexes showed enhanced antiproliferative activity against MCF-7 breast cancer cell lines. The Co(III) complexes of fluoro and bromo derivatives of ligands have displayed remarkable results with a roughly two-fold increase in their activity in correlation to the standard drug, Paclitaxel. Moreover, the fluorescence microscopy images of cells stained with acridine orange-ethidium bromide suggest an apoptotic mode of cell death [104].

### 3.5. Pd(II) Complexes of Thiosemicarbazides and Thiosemicarbazones

When a mixture of (*Z*)-2-(4-methoxybenzylidene)-*N*-phenylhydrazine-1-carbothioamide (**105**) with PdCl_2_ was refluxed in acetonitrile for 3 h, an orange precipitate of Pd-thiosemicarbazone complex **106** was formed in 75% yield (Figure 52) [105]. With Suzuki–Miyaura reactions under room temperature, via the addition of KPF_6_ and *N*-methylimidazole to acetonitrile and refluxing for 1.5 h, a yellow precipitate of complex **107** was obtained in 60% yield (Figure 52) [105]. The in vitro antioxidant activity showed that the Pd(II) complexes have effective antioxidant activities. According to the enzyme activity analyses, one complex (IC_50_ value 69.3 ± 5.2 μM) showed the most effective pancreatic lipase inhibition, whereas the other complex (IC_50_ value 25.7 ± 3.2 μM) had the most effective tyrosinase inhibition among the synthesized compounds [105].

The coordination geometry around the palladium is distorted square planar, with N(4)–Pd(1)–Cl(1), N(4)– Pd(1)–S(1), N(1)–Pd(1)–S(1) and N(1)–Pd(1)–Cl(1) bond angles at 87.72(15), 91.21(15), 84.11(15) and 97.18(14), respectively. The Pd1–N1, Pd1–N4, Pd1–S1 and Pd1–Cl1 bond distances are 2.015 A° (5), 2.019 A° (5), 2.2260 A° (18) and 2.3407 A° (18), respectively [105].

#### Utility of **107** in Cross-Coupling Reaction

To determine the catalytic activity of the precursor complex **107**, a model cross-coupling reaction was investigated at room temperature between 4-bromoanisole and phenylboronic acid. Screening various solvents using K_2_CO_3_ as the base in the presence of **106** showed that ethanol was the most effective solvent. The yield of 4-methoxy-1,1′-biphenyl under the condition described in Figure 52 was 93% yield [105].

Four Pd(II) complexes have been synthesized by reacting equimolar ratio of [PdCl_2_(PPh_3_)_2_] and 4(*N*)-substituted 4,6-dimethoxysalicylaldehyde thiosemicarbazone ligands **108** in toluene for 5 h (Figure 53) [106]. The orange crystals of palladium complexes **109a,b,** and **110a,b** were obtained in 63–72% yields (Figure 53) [106]. The substituents appear to affect the type of product formed to give either **109** or **110** (Figure 53). All the complexes’ antioxidant properties showed moderate activity compared to standard BHT. The cytotoxicity of the Pd(II) complexes was investigated in vitro against both lung cancer (A549) and human breast cancer (MCF-7) cell lines by using MTT assay and by using (AO/EB and DAPI) staining method for cytological changes in cell lines [106]. All the complexes inhibit the growth of the cancer cells significantly when compared to the standard. IC_50_ values of **109a,b,** and **110a,b** against breast cancer cells were calculated as 18 ± 1, 33 ± 1, 24 ± 1, and 28 ± 1 μM/mL for **109a,b,** and **110a,b** in MCF-7 cell lines, respectively. For A549, lung cancer cells, IC_50_ values were 22 ± 2, 25 ± 1, 15 ± 1, and 30 ± 2 μM/mL for **109a,b,** and **110a,b**, respectively. The observed IC_50_ values of the complexes prove moderate activity compared to the standard *Cisplatin* [106].

*Fluorescence spectral studies.* Fluorescence spectral studies have been widely used to study the interaction of small molecules with protein molecules. The interactions of BSA with **109a,b,** and **110a,b** were studied by fluorescence measurements at room temperature. The intensity of the fluorescence band of BSA at 349 nm was quenched to the extent of 69.6%, 75.7%, 40.9%, and 72.1% from its initial intensity upon the addition of **109a,b,** and **110a,b** with a hypsochromic shift of 5, 3, 5 and 3 nm due to formation of a palladium-thiosemicarbazide–BSA complex [106].

Similarly, the thiosemicarbazones of 4(*N*)-substituted 4,6-dimethoxysalicylaldehyde **111a**–**d** reacted with [PdCl_2_(AsPh_3_)_2_] with Pd(AsPh_3_)_2_, and two structures of palladium thiosemicarbazones **112a,b** and **113a,b** were obtained depending upon the type of substituent (Figure 54) [107]. All the complexes indicated their DNA/protein binding ability by using absorption and emission titrations. Investigations of antioxidant properties showed that all the complexes have significant radical scavenging properties. The anti-cancer activity of Pd(II) complexes was probed in vitro cytotoxicity against human breast (MCF7) and lung (A549) cancer cell lines by MTT assay. Further, AO/EB and DAPI staining methods were carried out to detect the cell death induced by the complexes. Complex **112a** exhibited better cytotoxic activity [107]. The palladium complexes **111a**–**d** were analysed using cell inhibition expressed in IC_50_ values and were found as 36 ± 1.0, 27 ± 1.0, 35 ± 1.0 and 29 ± 1.0 for MCF-7 cell lines. In the case of the A-549 cell line, the IC_50_ values were found to be 20 ± 1.5, 23 ± 1.5, 24 ± 1.5, and 23 ± 1 for complexes 1–4, respectively. All complexes showed moderate activity compared to the standard cisplatin. In addition, the coordination of the ligands to the Pd(II) ion enhances the antiproliferative activity of the complexes against both cell lines [107].

Crystallographic studies showed that complexes **112b** and **113a** distorted square planar geometry around palladium metal ions [107]. In complex **113a,** the palladium (II) ion is coordinated through the thiolate sulphur atom with Pd(1)-S(1) bond distance of 2.246 (5) Å and the nitrogen atom with a Pd(1)-N(1) bond distance of 2.063 (12) Å. The remaining sites are occupied by a chlorine atom with Pd(1)- Cl(1) bond distance of 2.337 (4) Å and triphenylarsine with a Pd(1)-As(1) bond distance of 2.355 (2) Å, respectively. The triphenylarsine in the complex N(1) nitrogen atom and thiolate sulphur (S1) chlorine atom (Cl) are mutually trans to each other. The trans angles (S(1)- Pd(1)-Cl(1) is 173.29 (17)° and N(1)-Pd(1)-As(1) is 175.8 (4)°) deviate considerably from the ideal angle of 180°, distorting the square planar geometry of the complex. The bond distances and angles between palladium and the coordinated atoms are similar to the reported Pd(II) complexes. In complex **113a**, a hydrogen bonding interaction is found with the donor-acceptor distance of 2.580 (17)° corresponding to the N(2)-O(1) [107].

The reaction of PdCl_2_ appended thiosemicarbazone ligands **114a**–**c** with PdCl_2_ (where X = 5-chloro (**114a**), 5-bromo (**114b**), and 5-nitro (**114c**)), and the palladium (II) complexes **115a**–**c** were obtained (Figure 55) [61]. The antimicrobial activity results observed that complexes **114a** and **114b** registered potent antibacterial activity against *B. subtilis* and *K. pneumoniae*, and complex **114b** showed good antifungal activity against the microorganisms. The antioxidant activity analysis revealed that complex **114c** showed significant activity with IC_50_ values 7.24 ± 0.09 µM. Moreover, the in vitro antiproliferative activity results suggested that complex **114c** exhibited high activity against the HeLa cell line compared with the standard with the IC_50_ value 16.52 ± 1.08 µM. The docking results correlate well [61].

Investigation of the antibacterial activity of steroidal thiosemicarbazone **116** and its tetradentate Pd(II) metal complex **117** was established by the reaction of the thiosemicarbazones with PdCl_2_ (Figure 56) [108]. The antibacterial activity of **117** was tested in vitro by the disk diffusion assay against two Gram-positive and two Gram-negative bacteria using cultures of *S. aureus*, *S. pyogenes*, *S. typhimurium*, and *E. coli*. Amoxicillin (30 μg) was used as the standard drug, whereas a DMSO-wetted disk was used as the negative control. The results showed that steroidal **116** is a better inhibitor of both types of bacteria (Gram-positive and Gram-negative) than steroidal thiosemicarbazone [108].

The reaction of formylferrocene thiosemicarbazone **118** with Pd(AcO)_2_ in an ethanol solution with a 1:2 ratio, and stirring for 24 h gave rise to heterometallic trinuclear complex **119** in 68% yield (Figure 57) [109]. Among other complexes (e.g., Ni and Zn), the Pd complex showed higher photocatalytic activity than the formylferrocene thiosemicarbazone free ligand **118**. Theoretical studies were used to characterize the compounds’ geometry and electronic structure and to provide a rational explanation for the measured [109]. The molecular structure of **119** displays a square planar geometry with a palladium (II) centre and two bidentate ligands, each of which coordinates to Pd (II) via the imine nitrogen atom and the thioamide sulphur atom of the deprotonated ferrocenyl thiosemicarbazone ligand. Moreover, both ferrocenyl groups in the complex are *anti* to each other, thus minimizing steric repulsion between them [109].

The palladium(II) complexes of indole-3-carbaldehyde thiosemicarbazones **121a**–**e** have been obtained by combining the ligands **120a**–**e** with [PdCl_2_(PPh_3_)_2_] in a 1:1 molar ratio (Figure 58) [36]. The reaction mixture was stirred for 4 h at room temperature to give the complexes 79–81% yields (Figure 58) [36]. X-ray structure analysis of the obtained complexes showed that the coordination geometry around palladium(II) could be described as distorted square planar; the palladium ion was bonded to the monobasic bidentate NS^-^donor ligand in such a way that a five-membered ring was formed, and the remaining sites were occupied by one chlorine and one triphenylphosphine [36]. The anti-cancer activity of the complexes was compared with that of the well-known anti-cancer drug cisplatin, and it was inferred that complex **121d** exhibited comparable activity. All the complexes displayed moderate anti-cancer activity against A549 and MCF7 cancer cell lines and less toxicity towards the normal cell line. The morphological changes assessed by staining methods and DNA fragmentation revealed that cell death occurred by apoptosis [36].

It has been reported that design and structure–activity studies using mononuclear palladium (II) complexes **123a**–**e** on a patient isolate of *Trichomonas vaginalis* are highly resistant to the FDA-approved drug metronidazole [110]. The synthetic approach was established by the addition of the appropriate thiosemicarbazone ligands **122a**–**e**, which were added to dry EtOH, under argon gas, to a mixture of triethylamine and Pd(PPh_3_)_2_Cl_2_ (Figure 59) [110]. Two compounds had similar IC_50_ values between the resistant strain and a previously analysed sensitive line. The most potent compound had an IC_50_ value of 15 μM on parasite growth and showed no effects on common normal flora bacteria or morphological effects when tested on cultured mammalian cells. The formed metal complex **123c** has been evaluated as catalyst precursors for the Mizoroki–Heck coupling reaction between a variety of electron-rich and electron-poor aryl halides and olefins to form the corresponding Chalcone (Figure 59) [110]. The palladium complexes (1 mol% loading) were found to catalyse these reactions effectively, with high yields obtained when aryl iodides and aryl bromides were utilized. The effects of the base, catalyst loading, reaction temperature, and reaction time on the catalytic activity of the most active complex were also investigated [110].

Palladium (II) complexes **125a**–**g** were prepared from the reaction of thiosemicarbazones (TSCN) **124a**–**g** with PdCl_2_, as shown in Figure 60 [111]. Coordination via the thionic sulphur and the azomethine nitrogen atom of the thiosemicarbazones to the metal ion was confirmed by spectral data. The TGA (under nitrogen, rate 10/min) profiles of complexes and the % weight at different temperatures were recorded. These complexes do not lose weight up to 245 °C. Further increments in temperature cause the decomposition of the complexes in two steps. The temperature range for the first step was 245–395 °C, where the loss of mixed fragments was observed. The second step starts immediately after the first one and continues until the complete decomposition of the ligand and formation of MS [M = Pd(II)] as the end product. The total% weight loss corresponds to the loss of the respective ligand after considering the transfer of one sulphur atom to the metal ion, and the residue corresponds to the metal sulphide [111].

Compounds **125a**–**g** were screened in vitro against the HK-9 strain of *Entamoeba histolytica*, which possesses amoebicidal properties. With enhancement of anti-amoebic activity, they showed fewer IC_50_ values than metronidazole. That resulted from the introduction of palladium metal in the thiosemicarbazone moiety [111]. All the complexes are more active than their respective ligands, indicating that the complexation to metal enhances the activity of the ligand. This may be explained by Tweedy’s theory [112], according to which chelation reduces the polarity of the central metal atom because of the partial sharing of its positive charge with the ligand, which favours the permeation of the complexes through the lipid layer of the cell membrane. The most active compounds in this class were again those thiosemicarbazone Pd(II) complexes which have cyclooctyl amine (**125f**, IC_50_ = 0.81 μM) [111].

Similarly, thiosemicarbazones were derived by thiophene **126a,b** formed with PdCl_2_(PPh_3_)_2_ in toluene followed by the addition of Et_3_N base, the corresponding tetradentate Pd complexes **127a,b** (Figure 61) [113]. The bond parameters illustrated by X-ray structure analysis confirmed the distorted square planar geometry around the palladium centre [113].

A series of 6-methoxy-2-oxo-1,2-dihydroquinoline-3-carbaldehyde 4*N*-substituted thiosemicarbazone ligands **128a**–**d** and their corresponding tetradentate palladium(II) complexes (i.e., PdCl_2_(PPh_3_)_2_) **129a**–**d** (Figure 62), was synthesized to evaluate the effect of terminal *N*-substitution in thiosemicarbazone moiety on coordination behaviour and biological activity [114]. The interactions of the new complexes with calf thymus DNA (CT-DNA) have been evaluated by absorption and ethidium bromide (EB) competitive studies, which revealed that complexes **129a–d** could interact with CT-DNA through intercalation. Further, the interactions of the complexes with bovine serum albumin (BSA) were also investigated using UV–visible, fluorescence, and synchronous fluorescence spectroscopic methods, which showed that the new complexes could bind strongly with BSA. Antioxidant studies showed that all the complexes have strong antioxidant activity against 2-2′-diphenyl-1-picrylhydrazyl (DPPH) radical and 2,2′-azino-3-ethylbenzthiazoline-6-sulfonic acid diammonium salt (ABTS) cation radical. In addition, in vitro cytotoxicity of the complexes against human lung cancer (A549) cell line was assayed, which showed that **129d** has higher cytotoxic activity than the rest of the complexes and cisplatin [114].

### 3.6. Ag(I) Complexes of Thiosemicarbazides and Thiosemicarbazones

4-Hydroxybenzaldehyde-thiosemicarbazone **130** reacted with AgNO_3_ and Ph_3_P in a molar ratio of 1:1:1, and the tetradentate silver(I) complex **131** was synthesized (Figure 63) [115]. The thermal behaviour of the complex was studied using thermogravimetry to evaluate its thermal stability and decomposition pathway [115].

The tetradentate silver(I) complexes **134a**–**c** containing 2-formylpyridine-*N*(4)-*R*-thiosemicarbazones **132** and 1,10-phenanthroline (phen) (**133**) were synthesized (Figure 64) [116]. In these complexes, phen and thiosemicarbazone ligands are coordinated in a chelating bidentate fashion. Compounds **134a–c** not only showed well in vitro antiproliferative activity against human lung (A549) and breast tumour cells (MDA-MB-231 and MCF-7), with IC_50_ values ranging from 1.49 to 20.90 μM but were also demonstrated to be less toxic towards human breast non-tumour cells (MCF-10A) [116]. X-ray structure analysis showed that compound **134a** contains a mononuclear Cu(II) centre, two bidentate bpy moieties, and a bidentate NO_3_− anion, resulting in distorted octahedral geometry [116].

Heteroleptic silver(I) complexes 137a–d were synthesized in good yields via the aerobic reaction of sodium naproxen **135** (1 mmol) with 2-(1-(4-substituted phenyl)ethylidene)hydrazinecarbothioamide (**136a**–**d**, 1 mmol) together with AgNO_3_ (1 mmol) in methanol as a solvent (Figure 65) [117]. The structure of the obtained complexes **137a**–**d** showed an asymmetric bidentate coordination mode of carboxyl oxygen atoms of naproxen with a silver(I) ion. The complexes are stable for 72 h, and biocompatibility was analysed towards normal human dermal fibroblast cells, which showed a non-toxic nature up to 100 ng/mL. In vitro antiproliferative activity of the complexes by MTT assay was tested against three human cancer cell lines and one non-tumorigenic human breast epithelial cell line (MCF-10a) in which the complex **137a**–**d** exhibited enhanced activity [117]. The cell viability decreased with increasing complex concentrations, showing the concentration-dependent nature and IC 50 values. Complex **137c** exhibits weak cell growth inhibition activity against all three cancer cell lines, and complex **137d** shows moderate activity towards the MCF-7 cell line compared to the standard drugs. The selectivity of complex **137d** towards MCF-7 compared to MDA-MB-231 and PANC-1 cell lines may be due to the electron-releasing substituent [117].

The approach dealt with the formation of six sulphur-bridged dinuclear silver(I) thiosemicarbazone complexes **138** (Figure 3) [118], which were synthesized through the reaction of silver(I) nitrate with 4-phenyl-3-thiosemicarbazone derivatives together with PPh_3_ in a 1:1:2 molar ratio. It was found that the thiosemicarbazone ligand exists as a thione rather than as a thiol tautomer. Subsequently, MDA-MB-231 and MCF-7 breast cancer cell lines and the HT-29 colon cancer cell lines were used to investigate these complexes’ in vitro antiproliferative activities. In all cases, the IC_50_ values were in the full micromolar range. Furthermore, the complexes had good anti-plasmodial activity against chloroquine-resistant *P. falciparum*, as per the results of histidine-rich protein 2 (HRP2) assays and cytotoxicity evaluations of MDBK cells [118].

Similarly, when Ph_3_P was replaced by diphenyl(*p*-tolyl)phosphine, the same structure of silver complexes **138** (Figure 4) was obtained and assigned as **139** [119]. The in vitro antiproliferative activity of these complexes was investigated towards the MDA-MB-231 and MCF-7 breast cancer cell lines, as well as the HT-29 colon cancer cell line, which yielded IC_50_ values in the low micromolar range. The antiplasmodial activity of these complexes was also examined against chloroquine-resistant *P. falciparum* parasite, which demonstrated good activity and was further tested for their selectivity index [119].

It was reported on the synthesis, characterization, and crystal structure of a mononuclear silver(I) tetradentate complex, [Ag(3-phenylpropenal-thiosemicarbazone)(PPh_3_)_2_]NO_3_ **141** (Figure 66) [120]. The complex was prepared by the reaction of thiosemicarbazone **140** and AgNO_3_ in the presence of PPh_3_. The ligand **140** was added to an acetonitrile suspension of AgNO_3_ and PPh_3_ (molar ratio, 1:2) and stirred for 0.5 h until a clear yellow solution was obtained. The solution was left at 4 °C for several days and then slowly evaporated at room temperature [120]. The minimum inhibitory concentrations (MICs) of the ligand **140** and its silver(I) complex **141** against two standard strains of Gram-positive (*S. aureus* ATCC-25923 and *E. faecalis* ATCC-29212) and Gram-negative (*E. coli* ATCC 25922 and P. aeruginosa ATCC-27853) bacteria showed that at 500 mg/mL, catsc has no antibacterial activity against any of the tested bacteria. The complex is inactive against *P. aeruginosa* at the 500 mg/mL concentration, but it is active against *S. aureus*, *E. faecalis*, and *E. coli*, with better activity against *S. aureus* than against *E. faecalis*. The considerably higher antibacterial activity of the complex compared with the free ligand, especially towards Gram-positive bacteria, has also been observed for the similar copper(I) thiosemicarbazone complex [120]. The differences in MICs found for the ligand and the complex are due to its ability to penetrate cell walls, which is structure dependent [120].

Silver(I) bromide complexes **143** containing triphenylphosphine (PPh_3_) and 4-phenyl-thiosemicarbazide (**142**) were prepared and structurally analysed, namely [AgCl(thiosemicarbazide)-(PPh_3_)_2_]CH_3_CN (Figure 67) [121]. Complex **143** exhibits a distorted tetrahedral metal coordination environment with two P atoms from two PPh_3_ ligands, one terminal S atom from the **142** ligands, and a chloride ion. The resulting reaction mixture was heated under reflux for 7 h. The resulting Ag complex **143** was obtained in 66% yield [121]. X-ray structure analysis showed the distorted tetrahedral metal coordination environment with two P atoms from two PPh_3_ ligands, one terminal S atom from the 4-PTSC ligand, and a chloride ion [121].

Silver(I) complexes **145a,b** were formed by the reaction of thiosemicarbazones derived by imidazole moiety **144a,b** with AgNO_3_. The reaction was carried out by stirring a methanol solution of the desired ligand **144a,b** with an aqueous solution of AgNO_3_. The reaction mixture was kept under stirring for 24 h. Silver complexes were obtained in 69 and 89% yields (Figure 68) [122]. The silver(I) complexes **145a,b** showed antifungal activity against *Candida* fungal strains, while the un-complexed **144a,b** ligands were inactive, suggesting that the antifungal effects are probably due to the presence of silver. All compounds exhibited potent antimicrobial effects against several aerobic bacterial strains, indicating that their mode of action probably involves an aerobic bio-reduction of the nitro group, with the formation of toxic metabolites [122].

## 4. Application of Thiosemicarbazone and Thiocarbazones Complexes of Transition Metals

### 4.1. Catalytic Applications

The use of thiosemicarbazones and other closely related chalcogen compounds as ligands in metal complexes has been a fruitful field of study for many years. However, initial reports on their application in catalysis did not appear until the 1990s [123], while their use in coupling reactions was only first reported several years later [124]. The versatility of metal complexes with compounds having hydrazinocarbothioamido groups can be exploited to develop new catalysts. Reactions that have been studied using these systems include oxidation [125], transfer hydrogenation [126], reduction [127], and various known reactions [128].

### 4.2. Biological Applications

As previously mentioned, thiosemicarbazones and thiocarbazones have shown widespread applications in medicinal and pharmaceutical fields. It was reported on the relationship between metal complexes in cancer therapy, highlighting some of these d-block properties of the corresponding metals [129]. The coordination of compounds with bonds between a central metal atom and surrounding ligands plays critical roles in biology, biochemistry, and medicine, controlling the structure and function of many enzymes and their metabolism [130].

The required general properties for metal complexes to exhibit potentially biological properties are:Adequately high thermodynamic stability to transport the metal to the active site;A good hydrolytically stability;A proper molecular weight. Low molecular weight compounds with no charge and very low water solubility have the advantage of being able to cross biological membranes by passive diffusion [130].

An interest in metals as antimicrobial and biocidal agents is reflected in hopes that they may prevent resistance [131,132]. Generally, various metal complexes have indicated greater biological activity than free organic compounds [133].

Metal ions’ biological activity depends on their concentration; they can either enhance or harm the organism’s health [134]. Metal chelate has been investigated to have several biological actions, including antibacterial, anti-fungicidal, antiviral, and anti-cancer activity [135,136]. The biological activities of metal complexes differed from those of free ligands or metal ions, and increased or decreased activities with the non-complexed semicarbazones/thiosemicarbazones have been reported for several transition metal complexes [137]. Thiosemicarbazones usually act as chelating agents with transition and non-transition metal ions connected through sulphur and nitrogen atoms and have immense pharmaceutical applications [138].

Lack of effective metal ions can cause many diseases, such as heart disease in very young children due to the deficiency of copper and pernicious anaemia due to cobalt deficiency. Natural detoxication is a mechanism of medicinal or physiological eradication of toxic substances from the human body, performed by the liver [139]. The latest research confirmed the upper anti-cancer effect of a thiosemicarbazone (TSC)–metal compound compared to TSC alone [140]. Some experimental evidence supported that the metal complexes of thiosemicarbazones usually show a higher bioactivity and lower side effects than free ligands [141].

## 5. Conclusions

In summary, it was here reported on the complexation of Cu(I), Cu(II), Co(II), Ni(II), Ag(I), and Pd(II) with ligands described as compounds containing a hydrazinocarbothioamido group. We also described the synthesis of these ligands. We also reported on different examples of synthesizing the formed complexes of the chosen metals with both thiosemicarbazides and thiosemicarbazones. Various modes of chelation occurred. Bidentate, tridentate and tetradentate complexes predominated. The metal complexes’ biological activities have shown to be more effective than the corresponding ligands, especially as antimicrobial agents. Since the ligation process of thiosemicarbazones and thiocarbazones ligands towards the assigned metals depends upon the type and structure of the ligands, different metal complexes with different coordination modes would be obtained. From the former point of view, studying the metal complexation of thiosemicarbazones and thiocarbazones would be valuable and would be chosen in various biological applications. Moreover, the metal complexes would increase the vitality of the field of catalysis.

## Data Availability

Not applicable.

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
