# Peer review of "Transition Metal Complexes of Thiosemicarbazides, Thiocarbohydrazides, and Their Corresponding Carbazones with Cu(I), Cu(II), Co(II), Ni(II), Pd(II), and Ag(I)—A Review"

_molecules, 2023, doi:10.3390/molecules28041808_

Round 1

Reviewer 1 Report (Previous Reviewer 1)

The author addressed most of my concerns in their revised manuscript. It may be published after some linguistic improvement and minor correction. I am still not satisfied with the organization and the items of this manuscript. I have never seen such long introduction. “Introduction” part should only focus on the background and the other part should be separated as independent items.

Author Response

Answer: According to the guides of reviewer 4, who drawn a new form of the introduction as:

  • Neglecting the numbering of the subtitles
  • Decrease the contents of the subtitles.
  • Change the numbering of titles
  • The applications of metal complexes were divided into two sections; one related to the catalytic and the other for the biological applications. Accordingly, paragraph including the biology in the introduction section dealt with biological applications of metal complexes was moved towards Section 4.
  • The introduction was reduced by removing some paragraphs (according to the advice of reviewer 4).
  • Refs were changed according to the aforementioned change
  • All changes were made in red highlights.
  • Thanks a lot

Reviewer 2 Report (Previous Reviewer 2)

The authors already have made improvements in the revised manuscript. However, there are still some major issues which must be address before acceptance. 1. Paragraphs 1 and 2 should be combined to illustrate the significance of transition metals. Besides, it is weird to describe “As an example, silver (Ag) metal has good antibacterial activity, and silver-based antimicrobials are better in terms of their effectiveness and non-toxic behavior to human cells [3]” in these paragraphs. This sentence should be explained in the sub-section of Ag, not in the “Introduction” section. 2. In the “Introduction” section, why did the authors only explain “thiocarbohydrazides”, not “thiosemicarbazides”. Both of them are shown in the title of this review, so they are equally important. 3. I have never seen that an “Introduction” section contains so many sub-sections. Actually, this review focused on “TM complexes bearing thiosemicarbazides”. I don’t understand why the authors used some many contents for the general introduction of transition metal catalysts (1.1-1.6), which is not highly related to the theme of this review. These contents should be concise and summarized into one or two paragraphs herein, and some related contents can be moved to the applications of these complexes (see the last sentence of this comment). Besides, section 1.7 should be moved to section 2 (2. Synthesis of thiosemicarbazides/thiocarbohydrazides as efficient ligands), while section 2 should be section 3 (3. Transition metal complexes ligated by thiosemicarbazides/ thiocarbohydrazides). Moreover, there is only one paragraph in “3. Thiosemicarbazone Complexes of Transition Metals as Catalysts”. I suggest to arrange this section into “4. Applications of thiosemicarbazide-/thiocarbohydrazide-containing transition metal complexes”, and divide it into several sub-sections such as “4.1 Catalytic applications”, “4.2 Biological applications”, etc. 4. In the “Conclusions” section, the authors mentioned “bidentate and tetradentate”, how about “tridentate”? 5. There are still so many grammatical, typo and formatting errors. I just pointed out several typical examples as follows, please read through the whole manuscript and correct all such mistakes. (1) Line 35, page 1: “have proved” should be modified into “have been proven”. (2) Line 41, page 1: What is the meaning of “Interestingly, metal complexes As an example,”? There must be some missing characters. (3) Line 42, page 1: “better” should be modified into “attractive”. (4) In Scheme 26: the dashed bond between “O” and “Cu” should be modified to an arrow-shaped bond. “CuCl2.2H2O” should be modified into “CuCl2·2H2O”

Author Response

The authors already have made improvements in the revised manuscript. However, there are still some major issues which must be address before acceptance.

  1. Paragraphs 1 and 2 should be combined to illustrate the significance of transition metals.

Answer: Was done

Besides, it is weird to describe “As an example, silver (Ag) metal has good antibacterial activity, and silver-based antimicrobials are better in terms of their effectiveness and non-toxic behavior to human cells [3]” in these paragraphs. This sentence should be explained in the sub-section of Ag, not in the “Introduction” section. 2.

Answer: Was done

In the “Introduction” section, why did the authors only explain “thiocarbohydrazides”, not “thiosemicarbazides”. Both of them are shown in the title of this review, so they are equally important.

Answer: Both have the same properties and were corrected as shown in the red highlights in the introduction. Moreover, in the title of the paragraph of metal complex as catalysis (before the conclusion section).

I have never seen that an “Introduction” section contains so many sub-sections. Actually, this review focused on “TM complexes bearing thiosemicarbazides”. I don’t understand why the authors used some many contents for the general introduction of transition metal catalysts (1.1-1.6), which is not highly related to the theme of this review. These contents should be concise and summarized into one or two paragraphs herein

Answer: The sections of the introduction were summarized. Numbering of sub-sections were removed

and some related contents can be moved to the applications of these complexes (see the last sentence of this comment).

Besides, section 1.7 should be moved to section 2 (2. Synthesis of thiosemicarbazides/thiocarbohydrazides as efficient ligands),

Was done

while section 2 should be section 3 (3. Transition metal complexes ligated by thiosemicarbazides/ thiocarbohydrazides).

Was done

Moreover, there is only one paragraph in “3. Thiosemicarbazone Complexes of Transition Metals as Catalysts”. I suggest to arrange this section into

“4. Applications of thiosemicarbazide-/thiocarbohydrazide-containing transition metal complexes”, and divide it into several sub-sections such as “4.1 Catalytic applications”, “4.2 Biological applications”, etc.  

Answer: Was done. However, the available biological applications were mentioned for each metal in Section 3. Anyhow the part includes the title Properties of metal complexes as biologically active compounds was shifted to be at the end of the review together with its refs thereby.

  1. In the “Conclusions” section, the authors mentioned “bidentate and tetradentate”, how about “tridentate”?

Answer: Was added

  1. There are still so many grammatical, typo and formatting errors. I just pointed out several typical examples as follows, please read through the whole manuscript and correct all such mistakes.

(1) Line 35, page 1: “have proved” should be modified into “have been proven”.

Was done

(2) Line 41, page 1: What is the meaning of “Interestingly, metal complexes

Was corrected (red highlights)

As an example,”?

Answer: Was corrected

There must be some missing characters. (3) Line 42, page 1: “better” should be modified into “attractive”.

Was done

(4) In Scheme 26: the dashed bond between “O” and “Cu” should be modified to an arrow-shaped bond.

Answer: Was corrected

“CuCl2.2H2O” should be modified into “CuCl2·2H2O

Reviewer 3 Report (Previous Reviewer 3)

The authors have made substantial changes to their manuscript; however, substantial errors remain. The electronic configuration of Ni(II) is not 4s23d8. Furthermore, these transition metal complexes should not be described as occupying the s orbitals when the d orbitals are not filled. This error is persistent within this review. 

The bonding configurations within complexes are still not consistent in their differentiation between covalent and coordination bonds. 

Author Response

The authors have made substantial changes to their manuscript; however, substantial errors remain. The electronic configuration of Ni(II) is not 4s23d8. Furthermore, these transition metal complexes should not be described as occupying the s orbitals when the d orbitals are not filled. This error is persistent within this review. 

Answer: I changed figure 1. The electronic configurations were done for the metal complexes of the assigned ions. The titles of the electronic configurations of the metals were also changed. According to the inquiry of the other reviewer the introduction section was also shortened (red highlights).

The bonding configurations within complexes are still not consistent in their differentiation between covalent and coordination bonds. 

Answer: I agree and they were corrected in the whole schemes

Reviewer 4 Report (Previous Reviewer 4)

I carefully read the revised Manuscript by Aly, Bräse at al.

I should stress out that the content, layout and English language now are significantly improved, but in general the text still requires the changing.

I did not notice a special generalization of the material presented.

Not all my comments were taken into account by the authors. Thus, l.570: “tridentate … atoms” or “{…}dentate metal complexes” (where “{…}dentate” means monodenate, bidentate etc.) are wrong. Complex may be “{…}coordinated” and only ligand may be “{…}dentate” (from latin “dentis”/”tooth”). These wrong Author’s submissions resulted to errors (for example, Scheme 14, where ligand 30a,b with two donor atoms, i.e. potentially bidentate, behaves as a monodentate ligand coordinated to Cu atom by S atom; tetracoordinated copper complex named wrongly as “bidentate”). The text contains a lot of such errors, and listing them by me will require a lot of my time, which, in principle, is not part of my duties. Furthermore, phrases like “tetradentate structures” (l.493; find other examples!) are also wrong. In the context of the Review complex or metal atom cannot be dentable.

Another my Comment was also ignored by the Authors. In Scheme 49 there is no cation.

Moreover, several Schemes contain mixed signs for bonding (dot lines, arrows…; i.e. Scheme 50).

Several noticed typos: ll. 37, 56, 327; 486 (are the molecules chiral?).

l.486: not “asymmetric”, but “nonsymmetric”.

Despite the requirement to the Authors to change the Manuscript only by Editing style, the Authors didn’t do so. The reference list was changed and is incorrectly formatted. How I can understand signs on l. 1170?

Author Response

I should stress out that the content, layout and English language now are significantly improved, but in general the text still requires the changing.

Answer: English language was polished

I did not notice a special generalization of the material presented.

According to the guides of reviewer 4, we did the followings:

  • Neglecting the numbering of the subtitles
  • Decrease the contents of the subtitles.
  • Change the numbering of titles
  • The applications of metal complexes were divided into two sections; one related to the catalytic and the other for the biological applications. Accordingly, paragraph including the biology in the introduction section dealt with biological applications of metal complexes was moved towards Section 4.
  • The introduction was reduced by removing some paragraphs (according to the advice of reviewer 4).
  • Refs were changed according to the aforementioned change

All changes were made in red highlights.

Not all my comments were taken into account by the authors. Thus, l.570: “tridentate … atoms” or “{…}dentate metal complexes” (where “{…}dentate” means monodenate, bidentate etc.) are wrong.

Complex may be “{…}coordinated” and only ligand may be “{…}dentate” (from latin “dentis”/”tooth”).

Answer: Thanks….That were changed in the whole MS (red highlight).

These wrong Author’s submissions resulted to errors (for example, Scheme 14, where ligand 30a,b with two donor atoms, i.e. potentially bidentate, behaves as a monodentate ligand coordinated to Cu atom by S atom; tetracoordinated copper complex named wrongly as “bidentate”).

Was corrected….thanks

The text contains a lot of such errors, and listing them by me will require a lot of my time, which, in principle, is not part of my duties. Furthermore, phrases like “tetradentate structures” (l.493; find other examples!) are also wrong. In the context of the Review complex or metal atom cannot be dentable.

Answer: The whole MS was revised

Another my Comment was also ignored by the Authors. In Scheme 49 there is no cation.

Answer: Was corrected

Moreover, several Schemes contain mixed signs for bonding (dot lines, arrows…; i.e. Scheme 50).

Answer: The bonding configurations within complexes were now in consistent in their differentiation between covalent and coordination bonds. 

Several noticed typos: ll. 37, 56, 327; 486 (are the molecules chiral?).

l.486: not “asymmetric”, but “nonsymmetric”.

Was corrected

Despite the requirement to the Authors to change the Manuscript only by Editing style, the Authors didn’t do so.

Answer: The style of the review was changed. The introduction was shortened. The numbering of subtitles were changed. The applications of metal complexes were divided into two sections; one related to the catalytic and the other for the biological applications. Accordingly, paragraph including the biology in the introduction section dealt with biological applications of metal complexes was moved towards Section 4.

The reference list was changed and is incorrectly formatted. How I can understand signs on l. 1170?

Was corrected. The refs were revised

Best regards

Ashraf

Round 2

Reviewer 2 Report (Previous Reviewer 2)

The authors have addressed all my concerns, so the manuscript can be published as the current form.

Author Response

Thanks good news

Reviewer 3 Report (Previous Reviewer 3)

Authors have made substantial edits to improve the quality of the manuscript. I recommend publication.

Author Response

Thanks

Reviewer 4 Report (Previous Reviewer 4)

Not all my Comments  are taken into account by the authors.

Author Response

Really i appreciate the concise reviewer coments

Really i reduced the introduction. I followed up the planned strategy of the other reviewers. The reviewer asked to reduce the introduction which i did.

So please if there is another thing which is special  i will directly do that. 

This manuscript is a resubmission of an earlier submission. The following is a list of the peer review reports and author responses from that submission.

Round 1

Reviewer 1 Report

In this manuscript, Ashraf et al tried to give a review of synthesis and biological activities of transition metal complexes of thiosemicarbazides, 2 thiocarbohydrazides, and their corresponding carbazones. It includes some development of bio-transition metal complexes research. However, I do not think it is sufficient enough for publication in Molecules as it is not well organized and there are too many errors and irregular writings. For examples, the items of this manuscript are badly organized and boring; I can not find figure 6 in the whole manuscript; some schemes are not being taken seriously; line 846, 885-894, why does the texts appear italic? Some comments and perspective from the authors themselves should be given in the conclusion.

Author Response

In this manuscript, Ashraf et al tried to give a review of synthesis and biological activities of transition metal complexes of thiosemicarbazides, 2 thiocarbohydrazides, and their corresponding carbazones. It includes some development of bio-transition metal complexes research. However, I do not think it is sufficient enough for publication in Molecules as it is not well organized and there are too many errors and irregular writings.

Answer: The English language was polished. The connection between sentences was made.

A short introduction (with its references in red highlights) was added to shed light on the titles of the review article. Rearrange of some parts was made (red highlights). Another important section Thiosemicarbazone Complexes of Transition Metals as Catalysts….as section 3. Its references were donated in redhighlight.

For examples, the items of this manuscript are badly organized and boring; I can not find figure 6 in the whole manuscript; some schemes are not being taken seriously; line 846, 885-894, why does the texts appear italic? Some comments and perspective from the authors themselves should be given in the conclusion.

Answer: Figure 6 was corrected; it was Figure 2. All the above were also corrected. The conclusion was also modified and declared. The paragraphs were effectively connected. A new title was added for the utility of thiosemicarbazones and thiocarbazones as catalysis (Item 3).

Reviewer 2 Report

See the attachment.

Author Response

In the paper's current state, I cannot recommend for publication, serious revisions are needed.

Answer: Many important paragraphs were added like those of x-ray structure analysis, Geometry of the complexes, utility of metal complexes as catalysts. Besides that introduction was added (All the former in red highlight)

In this review, the authors focused on some recent applications of transition metal complexes bearing thiosemicarbazides, thiocarbohydrazides, and their corresponding carbazones. They started the review with a description of the chosen five metals, including Cu [Cu(I), Cu(II)], Ni(II), Co(II), Pd(II), and Ag(I) and their electronic configurations. Then the authors group the synthesis methods of ligands and complexes. In my opinion, the total arrangement of this manuscript in not satisfactory.

Answer: A short introduction (with its references in red highlights) was added to shed light on the titles of the review article. Rearrange of some parts was made (red highlights). Another important section Thiosemicarbazone Complexes of Transition Metals as Catalysts….as section 3. Its references were donated in red highlight.

 It is just organized without careful consideration of scientific theory

Answer: I think that the scientific theory was declared through the presentation of electronic configuration and the ability of the assigned famous metals to coordinate with the hydrazinecarbothioamide group. Moreover, X-ray structure analyses of the formed metal complexes to declare their geometry was also added (red highlights). However, a few examples were also added (red highlight)

and electrochemical kinetics.

Answer: I am sorry. Studying and investigating metal complexes would include many items, including synthesis, biological activity, X-ray structure analysis, ……etc. I think these would be the most interesting. Moreover, few articles have dealt with the title mentioned above.

The authors only reviewed the synthesis methods of different ligands and complexes, but did not put forward the current challenges in the synthesis of such ligands and complexes, as well as the specific application prospects of the corresponding complexes. Besides, there are so many errors in the manuscript.

Answer: Corrections were made. English was polished. The connection between sentences was carried out. The biological application of organo-metal complexes was increased since it would be the most famous application of metal complexes. X-ray structure analysis, including the geometry of the formed complexes, was also illustrated (All the former in red highlight).

The authors should totally rearrange and re-write this review article so that it can be attractive and useful for researchers working in related areas.

This was done

The overall layout of the article is too chaotic, and there is no parallelism and progression among different sections. I also think that the author's proposal of thiosemicarbazones as ligands is too abrupt, without any background introduction.

Answer: That was previously explained by adding a brief introduction (red highlight).

It is farfetched to directly list the advantages of ligands and then use them to coordinate with various metals.

Moreover, the "Conclusion" section is too short. Please give a more detailed summary of the manuscript, point out the current challenges, and provide the future perspectives.

Conclusion was extended to include all the important parts in the review

  1. The presentation of compound structures and chemical formula is very unpleasant. For example, the errors in the following scheme (Scheme 26) were marked with red circles. There are many similar mistakes like these in this scheme, so the authors should thoroughly check the whole manuscript and modify all of them

Answer: It was corrected, and the others are similar.

There are serious graphical errors in Figure 1, which means that the authors are not professional at all. First of all, "S" should be "s". More significantly, the electronic configurations of Pd and Ag should be 4d 10 and 4d105s1 , respectively. Besides, the drawing of electronic configurations should be like the following figure.

Answer: It was corrected, and also those suggested by the other reviewers to write the lowercase s before d. Moreover, the titles of 4., 5.1., 5.2., 6.1. and 6.2. were corrected as well.

There are many typo and formatting errors in the text, such as "pd", superscript and subscript, spaces, etc. Please correct them carefully

Answer: It was corrected in the whole MS

In Chapter 7, there is only one subsection, so why is it divided into 7.1?

Answer: It was corrected (removed)

There are many typographical errors. For example, it was mentioned "The ability of metals to coordinate with ligands can be described as monodentate or bidentate (Figure 6)" on page 3. However, I went through the whole manuscript and could not find Figure

It was Figure 2 and was corrected.

Reviewer 3 Report

The authors describe a review on select late metals and related carbazone complexes reported with some detail regarding the biological activity. They focus on details regarding transition metal stability and electronic structure of generalized metal-ligand interactions followed by more indepth studies of specific metal examples. 

I would recommend that the authors take a very careful look at some of the broad generalizations made about transition metal complexes. For example:

"Transition metals can interact with various negatively charged molecules due to their various oxidation states". This is one of the reasons that explains transition metal behaviour, but is no way the only reason. For example, Ni(0) behaves vastly different when surrounded by ligands such as CO vs ligands such as PMe3. 

Furthermore, the authors made crucial mistakes regarding the electronic configuration of transition metal complexes. Indeed, in bulk metals the transition metals occupy the 4s orbitals prior to the 3d, but this is not the case in transition metal complexes. In any event, assigning Ni(II) as 4s23d8 is blatantly incorrect. This is just one example of the errors in electronic structure made. 

In addition, while I am not a supporter of the use of arrows to denote coordination bounds vs traditionally covalent bonds. One must be consistent! Complex 63 features a Co species various coordination interactions but has a charged phenolate donor. This should be listed as a non-arrow type of bond in the current model. Complex 145 features a silver species, where the direction of the arrows, denotes that the silver purely donates electron density to the electron rich triphenylphosphine ligands. 

In the paper's current state, I cannot recommend for publication, serious revisions are needed.

Author Response

I would recommend that the authors take a very careful look at some of the broad generalizations made about transition metal complexes. For example:

"Transition metals can interact with various negatively charged molecules due to their various oxidation states". This is one of the reasons that explains transition metal behaviour, but is no way the only reason. For example, Ni(0) behaves vastly different when surrounded by ligands such as CO vs ligands such as PMe3. 

Answer: I really appreciate the important comment of the reviewer. In this occasion, we added a paragraph in section 7.1. dealt with ligand definition. That would be a connected sentence to the reactivity of thiosemicarbazones as good ligands.

 Furthermore, the authors made crucial mistakes regarding the electronic configuration of transition metal complexes. Indeed, in bulk metals the transition metals occupy the 4s orbitals prior to the 3d, but this is not the case in transition metal complexes. In any event, assigning Ni(II) as 4s23d8 is blatantly incorrect. This is just one example of the errors in electronic structure made. 

Figure 1 was corrected and all the titles of  

In addition, while I am not a supporter of the use of arrows to denote coordination bounds vs traditionally covalent bonds. One must be consistent!

All bonds towards Metals were rechecked. However, using arrows to donate coordination bonds are widely used.

Complex 63 features a Co species various coordination interactions but has a charged phenolate donor. This should be listed as a non-arrow type of bond in the current model.

Was corrected

Complex 145 features a silver species, where the direction of the arrows, denotes that the silver purely donates electron density to the electron rich triphenylphosphine ligands. 

Was corrected

Reviewer 4 Report

The Review by Aly et al. concerns the synthesis of thiosemicarbazones, thiocarbohydrazones and related ligands, synthesis and description of several properties of metal (Cu+, Cu2+, Ni2+, Co2+, Pd2+, Ag+) complexes based on them. This subject represents interest to limited groups of scientists, but I believe that the subject may be reviewed.

I have serious problems reading the text; English language requires careful polishing. There are a lot of typos (for example, Figure 1), unsuccessful phrases (“biological aspects” (?) as a keyword; “copper complexes”; “numerous studies”, line 90, includes one (!) reference of 1997 year; “transition complexes”; “transition metals can interact…” etc.) and chemical errors (lines 403, 418: “bidentate complex”; Scheme 74, nitrato ligand; l. 713: there is no cross-coupling; l. 528: “tridentate O, N, S atoms” etc.), omissions (l.767: what bacteria; Schemes 37: Q; Scheme 49: cation etc.) and contradictions. It is definitely clear that the text was written by different people; there is no uniform style and editing. The text gives the impression of a compilation of different pieces; thus, for example, after Scheme 71 the similar data are presented in Figures 3, 4 with following Schemes 72 and so on. Furthermore, I didn’t see the generalization of literature data; thus, for example, the section concerning synthesis of ligands constructed by the principle of adding reactions (i.e. “… one more compound may be obtained”). As a serious disadvantage in metal complexes section should point out the lack of analysis of XRD data; related section containing general methods for analysis the features of these complexes is highly desirable. As a consequence of this the presentation of bonding in Schemes is contradictory (lines, dot lines, arrows…). The first sections (dedicated to metals) include evident statements (like “Co used to denote cobalt” etc.), what is not suitable for scientific publication, especially in “Molecules” (Q1 journal). Statements concerning the biological action of the complexes are also sketchy and very superficial (in many cases there are no IC50 data, mechanism of action etc.; l. 748: “better (?) cytotoxic activity”).

I didn’t understand why these complexes are biologically active? Which changes were observed under complexation (lines 64-65)? Better (?) coordination tendencies?

Conclusions are written very shortly and uninformative. I didn’t understand why these compounds are perspective/interesting, or why they are better and interesting than similar ones (like cisplatin or complexes based on other types of ligands etc.).

References should be abbreviated in accordance with https://cassi.cas.org/search.jsp.

I have a lot of Comments, but listing them would take too long time.

As a conclusion, I recommend rewriting this Review with subsequent additional submission.

Author Response

I have serious problems reading the text;

English language requires careful polishing.

Corrections were made. English was polished. The connection between sentences was carried out. The biological application of organo-metal complexes was increased since it would be the most famous application of metal complexes. X-ray structure analysis, including the geometry of the formed complexes, was also illustrated (All the former in red highlight).

There are a lot of typos (for example, Figure 1),

It was corrected

unsuccessful phrases (“biological aspects” (?) as a keyword; “copper complexes”;

It was corrected

 "numerous studies", line 90, includes one (!) reference of 1997 year;

The sentence was replaced by A review article showed that various metal complexes have indicated greater biological activity than free organic compounds [11]. The reference of 1997 was replaced by another review article in 2020 

"transition complexes";

It was corrected

"transition metals can interact…" etc.)

It was corrected into Generally, transition metals react

and chemical errors (lines 403,

Answer: It was corrected

418: “bidentate complex”;

Answer: was corrected to tetradentate

Scheme 74, nitrato ligand;

Answer: It was added and the structure of 158a,b was corrected

  1. 713: there is no cross-coupling;

Answer: was removed

  1. 528: "tridentate O, N, S atoms" etc.),

Was corrected

omissions (l.767: what bacteria;

It was added

Schemes 37: Q;

The results were shown

Scheme 49: cation etc.)

The assigned cations in the schemes were corrected

and contradictions. It is definitely clear that the text was written by different people; there is no uniform style and editing. The text gives the impression of a compilation of different pieces;

Answer: The English was polished

thus, for example, after Scheme 71 the similar data are presented in Figures 3, 4 with following Schemes 72 and so on.

Actually Refs 121 and 122. The same group of authors in different two publications. They did the same work and replaced one molecule of Ph3P with p-tolyl PPh3. In ref 123……the biological estimation was rewritten.

Furthermore, I didn't see the generalization of literature data; thus, for example, the section concerning synthesis of ligands constructed by the principle of adding reactions (i.e. "… one more compound may be obtained").

As a serious disadvantage in metal complexes section should point out the lack of analysis of XRD data; related section containing general methods for analysis the features of these complexes is highly desirable.

Answer: X-ray structure analyses of several metal complexes were added (red highlights).

As a consequence of this the presentation of bonding in Schemes is contradictory (lines, dot lines, arrows…). The first sections (dedicated to metals) include evident statements (like "Co used to denote cobalt" etc.), what is not suitable for scientific publication,

Answer: It was corrected

especially in "Molecules" (Q1 journal). Statements concerning the biological action of the complexes are also sketchy and very superficial (in many cases there are no IC50 data, mechanism of action etc.; l. 748: "better (?) cytotoxic activity").

Answer: That was also explained in many schemes (red highlight). The X-ray structure analysis was also extensively explained.

I didn't understand why these complexes are biologically active? Which changes were observed under complexation (lines 64-65)? Better (?) coordination tendencies?

It was corrected (red highlight)

Conclusions are written very shortly and uninformative. I didn't understand why these compounds are perspective/interesting, or why they are better and interesting than similar ones (like cisplatin or complexes based on other types of ligands etc.).

Conclusion was modified (red highlight)

References should be abbreviated in accordance with https://cassi.cas.org/search.jsp.

They were corrected according to the style of moloecules

As a conclusion, I recommend rewriting this Review with subsequent additional submission.

Answer: I did together with English polishing. All corrections were written in red highlights.